# Hidden Impact of Hardware Technologies on Throughput: a Case Study on a Brazilian Mobile Web Network

## Abstract

The Web has shifted towards a mobile-first ecosystem with tools, frameworks, and forums explicitly discussing and catering for the mobile users, both mobile apps and mobile web-pages. Unfortunately much of the studies and designs are often based on analysis and findings from developed regions (e.g., N. America and Europe) or based on user-generated data (introducing bias). In this paper, we present one of the first studies to understand the interplay between hardware characteristics (e.g., cellular and mobile) on expected network and application level performance in Brazil (the largest developing region in S. America). We analyze more than 170 million measurement sessions collected from within the network of one of the largest Mobile Network Operators in Brazil. Our findings (1) illustrate limitations of existing crowdsourced measurements and inaccuracies in assumptions about adoption patterns and performance in the global south, (2) highlight the differences between recommendations made by standardization bodies and real world performance, (3) disclose a significant change pre- and post-pandemic, and (4) quantify the benefits of using both client side and network data for analysis.

**Context and Scope.** We analyze how device and infrastructure factors impact mobile network performance in Brazil, emphasizing the need for data that reflects local conditions rather than relying on findings from developed regions. This is crucial for optimizing the mobile Web experience of users in diverse global contexts. We use a statistical ensemble to study network data from a mobile provider, identifying features that correlate with throughput and quantifying their impact. We use our dataset to identify, quantify, and detect differences between reality of the Brazilian infrastructure and standards/recommendations/crowdsourced measurements.

## 1 Introduction

Studies [7, 10, 24, 30, 44] project that the number of mobile users far exceeds nonmobile, e.g. desktop users, and that mobile connectivity is crucial to enabling upward mobility through digital services (studies show cell infrastructure improve GDP up to 12% in developing regions [41]). This importance has only skyrocketed with the COVID-19 pandemic and its acceleration of the adoption of digital services. Although the importance of mobile connectivity to the Web is undeniable, the designs, recommendations, and standardization often ignore the global south (e.g., S. America and Africa) and are defined by empirical characterizations of mobile performance and usage in developed countries [5, 21, 37]. Even worse, when they do include the global south they often use antiquated data. Moreover, many studies often focus on crowdsourced data which introduces several different types of sampling bias as

noted by others [33]. This bias is even greater in developing regions where users must pay for the data used for these measurements.

In short, there is a digital divide within network protocol designs and analysis due to the lack of representative data and the overheads associated with collecting these data that bias network analysis, protocol design, and standards. This bias is particularly problematic, as the mobile user experience is heavily affected by both the capabilities of the user devices and by the underlying infrastructure [27], thus designs on incorrect empirical data lead to significantly suboptimal performance.

In this work, we take a first step towards shedding light on these mismatches and closing the protocol digital divide by analyzing randomly-sampled measurements from of one of the largest mobile network operators in Brazil. We use a dataset with measurements collected over more than 7 million devices from 2023 and also 2020. Our dataset, and thus study, is distinguished from prior work [6, 11, 51] by several important factors: First, our dataset captures both pre- and post-pandemic time scales which allows us to understand the evolution of the performance in Brazil and quantify how the digital divide is changing. Second, our study is conducted using randomly collected sample across the country using carrier designed protocol which frees us from the bias of crowdsourced measurements. Third, our study captures data from both the client base and the cell network allowing us to make greater inferences and analysis than prior works which are limited to just one set (either client [53] or cell network [22]).

To conduct our study, we introduce a statistical ensemble which provides principled methods for understanding which variables are correlated with performance and estimating their impact in a manner that avoids conflating factors. Our pipeline also identifies meaningful changes in performance patterns. Our framework uses a combination of traditional statistical frameworks (e.g., Spearman) and machine learning (e.g, GradiantBoost peered with SHAP analysis) to identify correlation and estimate their impact. Our ensemble is tailored both to our domain and to our scale – for example, our use of covariate matching pairs to eliminate conflating factors hinges on the scale of our dataset, whereas the feature engineering for the GradientBoost is rooted in our domain (e.g., knowledge of the cell network characteristics).

Using our statistical ensemble we study the digital protocol divide along several dimensions: First, we estimate the impact of client and network variables on download throughput and perform an individual analysis of each feature to understand how it correlates with the outcome. Then, we perform a longitudinal analysis to quantify how the divide has changed over the pandemic in Brazil. Finally, we analyze the differences from the aggregated data available from Brazil (e.g., Speedtest) and from fine-grained data or data from the global north (N. America/Europe), both to highlight differences and potential biases.

*Conference'17, July 2017, Washington, DC, USA*
2024. ACM ISBN 978-x-xxxx-xxxx-x/YY/MM
https://doi.org/10.1145/nnnnnnn.nnnnnnn

**Table 1: Summary of results.**

| Ref | Findings and Descriptions | Implications |
|---|---|---|
| Tab 6 | In post-pandemic Brazil, the average mobile device is 25% faster with 75% more memory and a majority of them are now on 4G or 5G. | Post-pandemic Brazilian mobile ecosystem has leap-frogged ahead to be comparable to several European countries but still falls short of many other countries in the global north. |
| Fig 4b | A third of 4G (i.e., LTE) measurements recorded throughput similar to the top 18% of 3G (specifically, HSPA+) measurements | Connection generation does not predict performance, by itself, instead specific technology (e.g., HSPA+) and other features must be considered. |
| Fig 3, Fig 4c, Fig 5 | Client features (phone CPU clock speed) and internal MNO features (presence of hand over and base station load) are important metrics in understanding throughput. | Most datasets used for prediction and many prior studies on MNO performance often overlook one or more of these key features – thus, providing partially complete results. |
| Table 6 | Developer reference materials (e.g., Android or W3C recommendations) for mobile networks (technology and signal quality) are not aligned with current Brazil. | Developers require new reference standards to develop effective mobile apps and web pages – We provide an updated reference Table (Table 6). |
| Sec 8.3 | Crowd sourced throughput metrics (e.g., Speedtest) overestimate the mobile download average for Brazil due to sampling bias unique to the global south (e.g., per-byte data plans versus unlimited data plans). | Most open datasets for the developing regions are crowd sourced and thus over-estimate performance. Care must be taken when using these datasets to represent developing regions. |
| Fig 7a | Throughput degrades during non-business hours, especially from 6 to 10pm which is different from the after hour degradation observed by others [9] ( 4- 8pm). | This minor shift highlights a broader set of configuration changes that need to be made when porting operational and management tools to the global south. |

More broadly, our analysis has implications (listed in Table 1) for the operational aspect, diagnostic dimensions, and application/framework design of the mobile Internet. In particular, on the operational level, we observe a mismatch between usage patterns in Brazil and other countries, which implies a careful analysis of management frameworks before porting them to Brazil. Similarly, at the diagnostic level, we observe that traditional client active measurement methodologies are unable to capture crucial aspects of the cell infrastructure which impact performance. More broadly, crowd-source datasets and recommendations often used for application/protocol design are misaligned with current Brazil and will be unable to aide in design of techniques to further close the digital divide.

## 2 Related work

**Characterizing network impact on throughput:** Previous work on 5G [26, 50, 52] analyzed bandwidth and latency, identifying various forms of inefficiencies that lead to under-utilization of the technology optimal capacity. Our work differs in that we analyze data from a large-scale Brazilian deployment to understand network behavior.

In contrast, the significant body of work on 4G measurements [1, 8, 9, 14, 17, 32, 39, 47, 49, 51] highlights key challenges and characteristics of 4G networks generally in N. America or Europe. Although there is work that focuses on the Brazilian context [25, 35, 42, 43, 48], we differ from them in terms of scale and breadth: prior work has only focused exclusively on 4G or 3G or in specific technologies, whereas we explore three technologies at a larger scale over a longer period with access to richer performance data, thus allowing for broader characterizations and analysis.

There are a few studies on wireless network performance on different regions of the world. The most relevant previous studies that are related to our work [9, 16, 17, 22, 31, 32, 49, 53] are summarized in Appendix A. We observe that a fundamental difference is that our work is, to our knowledge, the first to present a comprehensive analysis of network variables and hardware characteristics (e.g., cellular and mobile) on expected network and application level performance using a real dataset obtained from one of the largest MNOs in Brazil.

**Ensuring quality of user experience:** Several works have studied various subsets of network variables and their implications for application performance [2, 4, 18, 20, 45, 48, 51]; we differ from prior work in several ways: first, we study many variables both from the cell provider and the client while prior work often focuses on a subset; second, we utilize measurements from a random sample of the MNO's entire client base; and third, we compare our results with other measurement methodologies and studies.

## 3 Background

Next we provide an overview of traditional cellular networks. A cellular or mobile network is one in which the end nodes are connected by a cellular wireless link. A set of physical base stations provide cellular connectivity to a geographical area called *cell*. Each base station can divide its radius into multiple cells, and it is common to have a base station that spans six cells in newer technologies. In addition, there are certain scenarios where cells from multiple base stations may overlap in a geographical location. To simplify our terminology, we refer to a single cell ID as corresponding to a single base station, even if in practical terms the physical structure may serve many cells.

Client connections use different technologies, such as LTE or New Radio, to communicate the device with the base station. Physical base stations can support multiple technologies, and are connected to the Mobile Network Operator (MNO)'s backbone network via optical fiber or radio links. The cellular network contains other components, e.g., the EPC, which we do not discuss because of a lack of direct relevance to our work. To provide context, our dataset (Section 4) comprises measurements taken from Android smartphones (connected to the MNO's cellular network) to measurement servers (colocated within the mobile's network through optic fiber cables). The measurement traffic does not pass through the public Internet, as both ends are inside the MNO's infrastructure. This is done to reduce the influence of the network path on the studied variables. An illustration of the system is at Fig 1.

## 4 Cellular Network and Dataset

Our data set consists of measurement sessions collected from 2023 (Jan-1 to Aug-30) and 2020 (Jan 1-31), from one of the largest mobile

**Table 2: Summary of the dataset.**

| Devices | Base station cells | Brands | Models | Measurements |
|---|---|---|---|---|
| 7,606,173 | 860,799 | 307 | 4,292 | 176 million |

**Table 3: Technology distribution, per million (M) of samples.**

| 5G | 4G | | 3G | | | | 2G |
|---|---|---|---|---|---|---|---|
| NR | LTE_CA | LTE | HSPA+ | HSUPA | HSPA | UMTS | EDGE |
| 1.7 M | 32 M | 122 M | 17 M | 1.4 M | 1.3 M | 0.14 M | 0.017 M |

network operators (MNO) in Brazil. The data was collected via an active measurementas system which is briefly described below.

**Measurement system**: The system (Figure 1) comprises three modules. The *agent* which is a software package, i.e, MNO Android App, that runs on the users cell phone and periodically wakes-up to run a set of active measurements against a dedicated server while simultaneously reporting hardware metrics about the cellphone and the base station. The *MS* which is a measurement server that acts as an endpoint and reflects the measurement traffic sent by an agent. In Brazil, the MNO deploys approximately one MS per state, or multiple MSes for regions with high population density. Finally, the MNO maintains a storage infrastructure, called the *Admin* module, which stores the measurement data and provides a UI for the administrator to control the system and monitor performance indicators.

A measurement data point is collected using the following workflow: first, the agent requests a test agenda to the Admin module. The agent then sends a request to the MS containing the parameters for the test. The MS sends a burst of measurement data to the agent. After receiving it, the agent will respond with symmetric traffic to test the upload channel and send the result metrics for the first burst. Upon receiving the entire reply, the MS can compute the resulting metrics of the session and report it back to the admin. The system implements the guidelines for active IP measurements established by the IETF's IP Performance Metrics (IPPM) RFC [13].

**Collected data:** We collected 176 million measurement data points from more than 7 million unique devices (Table 2 and Table 3). The selection of devices follows a proportional stratified random sampling logic, aiming to maximize the geographic coverage of measurements, and to carry out more measurements in regions with high population density. Each measurement data point consists of

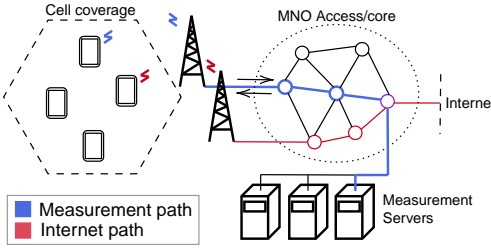

**Figure 1: Infrastructure of an agent on a cellular network connected to a measurement server. The measurement session path is entirely within the MNO network.**

a timestamp, the state of the connection during the session, the state of the device during the session, measured download, upload, latency and other metrics outside the scope of this study.

**Ethical and privacy concerns:** The data presented in this paper do not expose sensitive information about the subscriber's device and any characteristic or identifier that could otherwise be used for fingerprinting is anonymized. Additionally, the system does not analyze, collect nor has access to any user traffic that occurs outside of the performance measurement scope. The ISP provides its customers with an optional Android app[1] that allows them to manage their subscription and the measurement tool is included in this app. The use of data for measurement purposes is written in Terms and Conditions.

## 5 Statistical Ensemble

Our goals are (1) to understand and empirically identify the attributes that correlate with network performance, and (2) to identify the fundamental differences behind the characteristic of cellular networks in Brazil compared to widely available aggregate results. In this analysis, a crucial challenge is to ensure that there is a direct correlation due to the attributes being analyzed and not due to hidden attributes or factors not captured by our measurement system. Given that our measurements are performed in an uncontrolled environment, not only do all variables studied act simultaneously, but the result may also be influenced by other hidden variables and biases unknown to our dataset. Figure 2 presents ours statistical ensemble which consist of methods to identify features that are correlated with performance, to predict their impact, and to analyzing them over time to indicate fundamental changes in their behavior.

**Correlation Analysis (§ 6.2-§ 6.6):** In performing correlation, our goal is to identify the set of features on which are correlated with throughput while avoiding confounding features (recall, confounding features are features other than the independent features that are potentially associated with the outcome (dependent) feature).

We select our statistical method for correlation by identifying method that match the properties of our dataset. Namely, the relationships between the variables and the outcome which is not necessarily linear or independent, and that we deal with either continuous or ordinal data. Given these properties, we select, Spearman's Rank, a non-parametric test for our analysis over others (e.g. Pearson, Phi, Cramer's V).

To account for confounding features, we take advantage of the size of our data set and select the Matched pairs experimental design over alternative designs such as Propensity Score Matching and Inverse Probability Weighting. Matched Pair design requires few assumptions about the relationship between confounding features and outcome, however it is heavily dependent on the size of the dataset to find high-quality matched pairs. Specifically, Matched pair works by dividing the dataset in two groups, *A* and *B*, then matching each point in *A* with a point in *B* that has similar confounding values. This results in a group of pairs, such that each pair has similar characteristics with the exception of the independent variable we are interested in studying. Therefore, confounding are

---

[1]Apple devices are not considered in this study because the MNO's measurement system is based on Android.

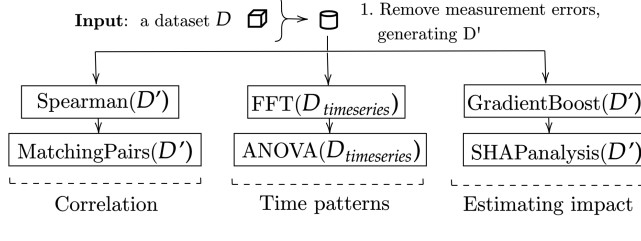

Input: a dataset $D$

1. Remove measurement errors, generating D'

Spearman($D'$)

MatchingPairs($D'$)

FFT($D_{timeseries}$)

ANOVA($D_{timeseries}$)

GradientBoost($D'$)

SHAPanalysis($D'$)

Correlation          Time patterns          Estimating impact

**Figure 2: Statistical Ensemble.**

neutralized and we can better understand the effect of the independent variable on the throughput. We match data points close to each other according to the Euclidean distance. With a pair in hands, we can compute the difference between the independent variable of the two data points: $\Delta X_i = X_{iA} - X_{iB}$ and likewise for our dependent variable $\Delta Y_i = Y_{iA} - Y_{iB}$. These differences can then be used to compute correlation coefficients.

**Estimating Impact (of various properties) (§ 6.1):** We estimate the impact of each variable by training an ML prediction task for throughput and then studying the prediction with SHAP analysis [29], a technique that computes the average contribution of each variable to the predicted value. Specifically, we train a regression task on a Gradient Boosting model to predict download throughput given network and device variables as input. The set of best-performing hyperparameters was chosen applying Grid Search to a subset of training data. The resulting model has an R-squared value of 0.233 and a Cross-Validation Root mean square error of 15.8 with standard deviation of 0.4. We expect the mean square error to be high as throughput has a lot of variance even accounting for all our input set variables, nevertheless this performance is enough to capture the effects of the feature set on the output value.

**Time Series Analysis (§ 6.7):** We also apply Fourier Transform for time series data with multiple decreasing or increasing patterns in the same time segment. Fourier Transform decomposes the underlying components of a time series into a series of sine waves. By analyzing the frequency and amplitude of those sine waves we identify recurring patterns in our data, and use ANOVA test to determine if these patterns are statistically significant.

## 6 Understanding Network Performance

We first estimate the impact that each feature has on throughput (§ 6.1), and then analyze individually the feature set to understand their correlation with download throughput (§ 6.2-§ 6.7), summarized in Table 4. We note that while we performed a time series analysis of the different features, we only found significant results in the "hour" feature which we expand on this analysis in § 6.7. The more fundamental longitudinal analysis of the other features is in the post and pre-pandemic differences (§ 7).

### 6.1 Estimating Feature Impact

To measure the impact of each variable on download throughput, we employ our model interpretability technique (Section 5) on the subset of features in our dataset that are statistically correlated with throughput, specifically: CPU clock speed, memory size and utilization, battery level, signal strength, connection technology,

handovers, number of devices under the base station and time of day.

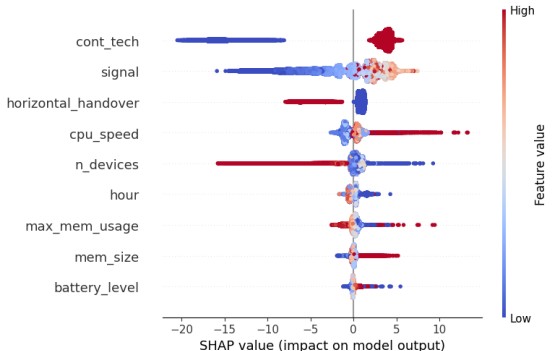

**Figure 3: Feature set ranked (top to bottom) by decreasing feature importance to the output value.**

Figure 3 presents the results of our analysis with variables ranked by impact. A "SHAP value" is the impact of a variable on the output (measured by the unit of the output, in our case it is throughput in Mbps) for a given datapoint. Each point in the figure shows the impact of a single datapoint. The main contributors to the prediction are Connection technology and signal with a mean impact of 6 and 4 Mbps, respectively, which is unsurprising because connection and signal both determine the performance range (as we show shortly). Next is handover with an impact of 1.5 Mbps, which creates a disruption in the connection. We see that, when a handover happens, the impact on throughput can be anywhere from -2 to -9 Mbps.

Next is CPU clock speed, a client-side metric, which has an impact of 0.97. We note that CPU clock speed has a high impact regardless of the cellular technology. Naturally, CPU speed impacts an application's ability to fetch and process data, and is correlated with other hardware characteristics (Appendix D). This has a broader impact for interpreting measurement results. Interestingly, memory has significantly less impact – on average, memory has a SHAP value of 0.22.

*Takeaway:* Our analysis further reinforces the role of hidden factors in introducing bias on online crowdsourced and client-side data available for analysis. In particular, we quantify the impact of bias for handover and we show that client-side information (e.g, device CPU), which is often not exposed, is just as important as network level metrics.

### 6.2 Connection Technology

We begin by analyzing cellular connectivity technology (or generation) and unsurprisingly it has a major impact on throughput. In Figure 4a we observe that on average, each generation has 4 times the download speed of the previous generation – 4G is 4 times faster than 3G, and 5G is 4 times faster than 4G. We note that the shape of the 5G violin in Fig 4a differs from the others: in 5G, a large portion of measurements recorded speeds closer to the peak of 550 Mbps, while the peak of previous generations is far from the bulk of measurements. For upload speed, the biggest improvement is from 3 to 4G connections, as it went from 1.7 Mbps average throughput with HSPA+ (3G) to 8.9 Mbps on LTE and 14

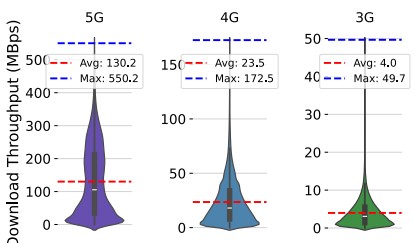

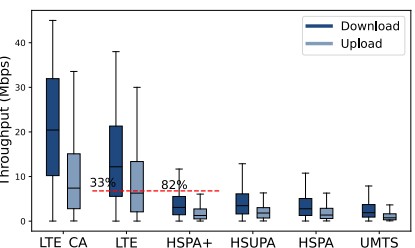

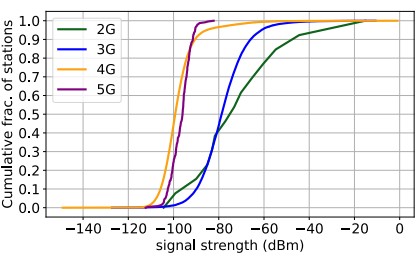

(a) Download Throughput per Generation     (b) Download and Upload per Conn Tech     (c) CDF of Avg. Signal Strength by Station

**Figure 4: Overview of throughput and signal strength under different network settings.**

**Table 4: Spearman's correlation coefficients for unmatched and matched pairs by feature and throughput.**

| Features | Correlation | | Matched Pair | |
|---|---|---|---|---|
| | Download | Upload | Download | Upload |
| Connection Technology | +0.47 | +0.43 | +0.37 | +0.38 |
| Signal Strength | -0.19 | -0.19 | +0.17 | +0.24 |
| Base Station Load | -0.29 | -0.22 | -0.17 | -0.10 |
| CPU Clock Speed | +0.12 | +0.06 | +0.03 | +0.02 |
| Memory Size | +0.15 | +0.05 | +0.04 | +0.01 |
| Memory Utilization | -0.03 | -0.01 | -0.02 | -0.03 |
| Battery Level | +0.005 | +0.005 | +0.03 | +0.02 |

Mbps on LTE with Carrier Aggregation (4G). From 4 to 5G it went from 14 Mbps to 36.5 Mbps on New Radio (5G).

These values can vary a lot. The standard deviation for LTE download speed is 13.32 Mbps, while the standard deviation for UMTS and HSUPA are 4.06 and 3.74. 3G and 4G can vary so much that the bottom third of 4G measurements have worse (< 6.8 Mbps) recorded download throughput than the top 18% percent of 3G measurements, as shown in Figure 4b. Upload throughput exhibits similar relationship. That is, in certain situations a 3G connection can record better throughput than a 4G connection.

We order each unique connection technology from oldest no newest and assign an integer to each position[2]. From now on, we always define the null hypothesis $H_0$ as that there is no relationship between the two variables. Spearman's rank coefficient (Table 4 – row 1) reports a positive correlation and rejects the null hypothesis thus statistically supporting our claim about the correlation between throughput and connection type. When we apply Matched pair (Table 4 – row 1) to account for confounding variables, we observe that the correlation is a little weaker however this is not surprising because there are implicit dependencies (for example, devices without LTE capabilities likely have smaller RAM and slower CPUs).

*Takeaway: Newer connection generations are around 4 times faster than previous generations. However, 4G connections can exhibit similar throughput to 3G connections quite frequently. About a third of measured 4G connections had a throughput similar to 3G averages and were slower than the fastest 18% of 3G connections.*

---

[2]Computing correlation coefficients require our variables to be numeric. As such, we use Ordinal Encoding to transform each connection technology into a numeric value.

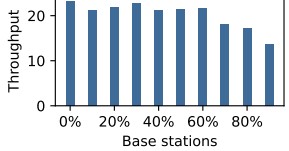

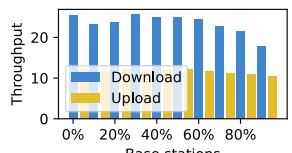

(a) Average download throughput per base station decile     (b) Average 4G download and upload throughput

**Figure 5: Throughput (Mbps) per base station decile.**

### 6.3 Signal Strength

In our dataset, signal strength is the device signal strength value during the measurement session. It is measured by dBm (decibels (dB) relative to 1 milliwatt (mW)). Values range from -120 dBM to 0. High dBm values are usually labeled as excellent signal strength, lower values often trigger the OS to change to a slower connection. The average signal strength in the dataset is -93 dBm, with a standard deviation of 13.5.

The correlation (Table 4 – row 2) between signal strength and download throughput shows an interesting results: it reports a negative correlation and indicates a counter intuitive results that a worse signal is correlated with a better throughput. This is where matched pairs prove their usefulness. When we apply Matched pair (Table 4 row 2), we observe that the correlation becomes positive. That is, when accounting for confounding features, a higher signal strength is correlated with a better throughput. Figure 4c helps understanding why the correlation changed when accounting for confounding features. From this figure, we observe that faster generations work on lower signal strength, despite having better throughput. When this characteristic goes unnoticed, it may seem that low signal strength itself results in better throughput.

*Takeaway: Signal strength has a positive correlation with throughput. Each generation has a distinct optimal dBm range, so comparing dBm values introduces bias in correlations if the connection generation covariate is not neutralized.*

### 6.4 Base Station Load (# of Attached Devices)

We use the number of devices measured under a cell ID on the same month as a proxy for traffic load. This number does not necessarily mean the devices were active or measuring at the same time on that base station, but instead that they were active within a 30-day window on the same base station. The average number of devices under a cell ID is 21. However, since most measurements happen on

cell IDs with many devices, this means that a measurement session takes place on a cell ID that has 116 devices connected to it on average and the standard deviation is 34.63.

In analyzing the correlation (Table 4 (row 3), we observe that download throughput has a strong correlation with load while upload through has a much weaker correlation. We observe that this correlations decreases slightly while account for confounding factors, it remains strong. A possible explanation would be that devices in urban centers use faster connection technologies, have better signal strength or are newer phone models.

Next, in Figure 5a, we try to analyze the how base station load impacts network performance. We observe that the top 20% of loaded base stations (top 2 deciles) have 81 devices on average, and record 15.3 Mbps of average download throughput. The bottom 20% of loaded base stations (bottom 2 deciles) have 11.2 devices on average and record 23.5 Mbps average download throughput. This is an increase of 53% in throughput compared to the top loaded base stations. We also note that the effect of load on throughput seems to be stronger on download throughput. In Figure 5b, while both download and upload decreases when the number of devices increase, the average download throughput decreases much faster than upload. This indicates that the overload caused by the number of devices is disproportionately affecting the downlink, likely due to client devices downloading much more data than uploading it, therefore increasing interference that results in packet loss.

*Takeaway: The number of devices under a base station is negatively correlated with throughput. In the worst case scenario, a crowded base station records almost half of the throughput measured on under-utilized base stations.*

## 6.5 CPU clock speed and memory

We now analyse the correlation between throughput and the phone's hardware specs (Table 4 rows 4-7). First, we analyze CPU and observe a small but noticeable correlation which decreases when we account for confounding factors. Next, we analyze maximum memory utilization (in % of total memory), we observe an insignificant monotonic relationship. Eliminating confounding features influence suggests that CPU speed on throughput (identified in § 6.1) is probably a result of its correlation with other hardware metrics, e.g., newer phone models with faster CPU speeds have a better network interface or are owned by people who have access to faster connection technology.

*Takeaway: CPU speed and memory size are positively correlated with throughput. Memory utilization shows no significant monotonic correlation to throughput.*

## 6.6 Horizontal Handover

We now study the horizontal handover effect on download and upload speed. Horizontal handover is the process of transferring a data session from one cell to another, without a change in the session's connection technology. When a handover does not happen throughout the measurement session, the average download and upload speed are 24 Mbps and 11.9 Mbps, respectively. When a handover happens during the session these values drop to 18.3 Mbps of download speed and 10.5 Mbps of upload speed, a decrease of 26% and 12%, respectively. That makes the presence of handover one

of the top indicators of throughput degradation from all variables considered. The decrease in download throughput reaches averages of 30% on 4G and 5G connections, while 3G connections are less affected by it with an average of 10%.

*Takeaway: Handovers greatly reduce throughput on average 26% for download and 12% for upload.*

## 6.7 Time influence on throughput

Each data point in the dataset contains a timestamp of the moment the measurement session started. The measurements were carried between 10AM and 10PM. Now, we analyze time of day to identify patterns that could affect throughput. Figure 7a shows a 1-minute average of download throughput, for all minutes between 10AM and 10PM. The averages seem to follow a pattern throughout the day. Earlier in the morning it stays between 22 and 24 Mbps. From 11:30pm to 1:30am it decreases to 20 Mbps at its lowest point. In the afternoon, it rises back closer to 23 Mbps and falls again after 5pm. These timestamps follow business hours in Brazil: from 8am to 12pm is the first shift, from 12pm to 1:30pm a lunch break and the second shift goes from 1:30pm to 6pm. The throughput decrease is similar to a study [9] in the US but shifted two hours later (11am vs 1pm).

We would like to determine if this pattern is statistically significant. We can infer from Figure 7a that the variables are not monotonic or linearly correlated, so we should avoid using Spearman correlation coefficient. We use ANOVA, a test to determine if there are any statistically significant differences between the means of independent groups.

We divide the day into four sections: morning (10am-12pm), lunch (12pm-1:30pm), second shift (1:30pm-6pm) and night (6pm-10pm). The null hypothesis $H_0$ is that there is no significant difference between the mean of each section. We perform the ANOVA test for these sections. The result is an $F = +452.13$ and $p = 10e-314$. The P-Value is extremely low, allowing us to reject the null hypothesis and consider that the means in fact differ above random chance. This pattern does not appear if we filter the data to include only weekends, which implies this effect is a consequence of workday schedule.

Having investigated the daily changes in throughput, we can use a Fourier Transform to identify other recurring patterns in our data. By decomposing the daily average throughput from January to August of 2023 into a series of frequencies, we analyze the resulting positive power spectrum. We found a peak closer to 0 Hz, which indicates the download throughput does not changes drastically in the 8-month period, as expected. We have another peak at 0.14 (or 1/7), which indicates that there is a recurring pattern on a weekly basis. We plot this pattern in Fig 7b. We can see that the throughput is slightly higher during the weekends, and decreases 8% in weekdays to an average of about 23.1 Mbps.

*Takeaway: Throughput is affected by daily and weekly patterns related to workday schedule.*

## 7 Post-Pandemic Evolution

Using our 2020 dataset, we perform a pre and post-pandemic analysis of the infrastructure and the clients. In particular, we aim to quantify how the divide has changed post-pandemic.

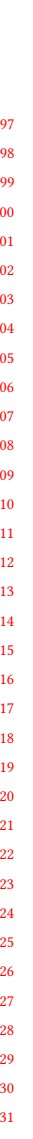

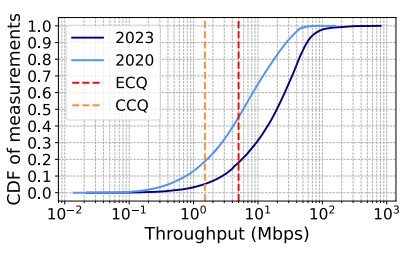
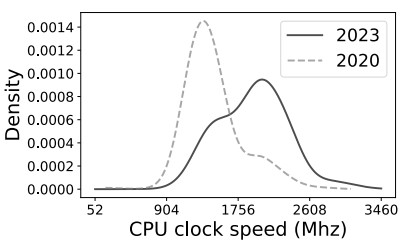
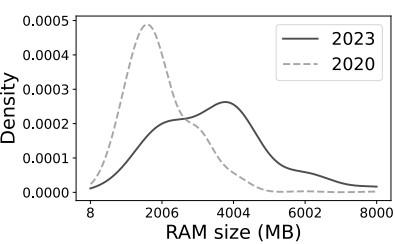

**(a) CDF of download throughput**  **(b) Device CPU clock speed**  **(c) Device RAM size**

**Figure 6: Comparison of pre (2020) and post-pandemic (2023) metrics. ECQ refers to Excellent Consistent Quality and CCQ to Core Consistent Quality, thresholds from [46].**

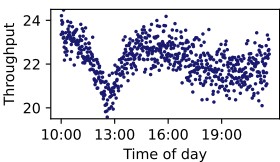
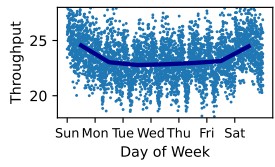

**(a) Daily Download Throughput**  **(b) Sunday to Saturday**

**Figure 7: 1-minute throughput (Mbps) averages throughout different time periods. A line represents the daily average.**

**Infrastructure Changes:** The network underwent several significant changes compared to January 2020: first, the adoption and implementation of 5G. Second, a shift of about 35.4% connections from 3G to 4G. Third, 2G went from a very small share of 1% in 2020 to a non-significant percentage in 2023. Finally, despite significantly more connections using 4G, we observed major improvements in 4G (65% improvement in throughput) due to the implementation of the LTE Carrier Aggregate (as confirmed by the operator). In Fig 6a we see the distribution of throughput in the pre- and post-pandemic datasets. We note that in 2020, 20% of measurements failed to meet the download throughput threshold for core web use-cases and 46% for emerging use-cases (e.g. live video streaming). In 2023, these percentages dropped to 5% and 16%.

**Client-Side Changes:** The dataset contains 4,292 phone models from around 300 smartphone brands. In analyzing hardware characteristics of these devices, we find that the average CPU clock speed went from 1.4 to 1.75 Ghz (as shown in Fig 6b) and RAM size went from 1.94 GB to 3.4 GB (Fig 6c). 48% of devices have at least 4 GB of RAM, against the global percentage of 66% according to a mobile market study [36]. In terms of connectivity, in 2020 the percentage of devices capable of 5G was negligible, whereas in 2023 10% of devices supported 5G.

*Takeaway:* Taken together, we observe significant improvement in the mobile web ecosystem (MNO infrastructure and end user mobile devices) and its ability to support emerging applications.

## 8 Performance Comparison

In this section, we compare our results with MNOs in the global north to quantify the digital divide (§ 8.1), with other MNOs in Brazil to illustrate generality at the aggregate level (§ 8.2), with crowdsourced results to illustrate unrepresentativeness of their

data (§ 8.3), and then conclude with the limitations of standards and recommendations (§ 8.4).

### 8.1 Characterizing the Global Digitial Divide

Next, we compare against recent studies using test devices across 8 European and American cities [23] reported uplink throughput values for 4 and 5G connections. While not as extensive as ours, a comparison allows us to understand broad differences. Our comparison shows that modern Brazilian networks[3] are comparable to cities such as Madrid and Porto and Bay Area, while falling behind Berlin, Oslo and Turin metrics. A country-wide 5G study on the US with test devices [31] reported significantly (about 50%) higher 5G NSA downlink performance than our study.

Comparable study [38] on 4G (LTE) performance in the global south using test devices in Malaysia, Singapore, and Thailand reported average video download speed ranging from 1.9 to 9.8 Mbps, which is about 63% lower than the averages reported on our dataset.

*Takeaway:* We observe that post-pandemic Brazil shows remarkable performance compared to other areas in the global south (e.g., Asia-Pacific) and comparable to some Western European cities.

### 8.2 Other performance studies in Brazil

We compare our data against OpenSignal's measurements study of 4G and 5G across all major Brazilian MNOs in Brazil [34]. Our data, from the first half of 2023, shows a significant improvement in 5G metrics and a slight improvement in 4G metrics, as Table 5 illustrates. The comparison of our dataset with the local Brazilian averages shows that our data (coming from one of the largest local MNOs) has the performance at least equal to the average of all Brazilian MNOs. However, this performance is still significantly slower than what speedtest platforms report for the country, as we detail in the next section.

### 8.3 Analyzing Bias in Crowdsourced Measurements

Next, we compare our data against a popular dataset for network analysis – Ookla. The Ookla speedtest platform publishes mobile speed averages for each country [40] which are based on crowd-sourced measurements. We compare the results of our study with their data on Brazilian Internet performance from March to August

---

[3]The values shown in our study (a median/75th LTE upload of 9/19 Mbps and 5G upload of 24/55 Mbps).

**Table 5: Network metric averages from OpenSignal 2022 [34]**

| Throughput (Mbps) | OpenSignal Global Top 41 (2022) | OpenSignal Brazil (2022) | Our study (2023) |
|---|---|---|---|
| 4G Download | 39.6 | 21.6 | 23.5 |
| 4G Upload | 11.4 | 8.5 | 12.3 |
| 5G Download | 187.1 | 51.7 | 130.5 |
| 5G Upload Speed | 25.1 | 18.3 | 36.5 |

of 2023. The comparison between our dataset and the averages for Brazil shows that upload values are roughly the same.[4] However, we observe some difference for downloads: the average download speed for all mobile technologies recorded by our data is 18 Mbps and the speedtest results show 43.03 Mbps, more than two times higher. These differences can be attributed to bias in the nature of crowdsourced measurements as highlighted by prior works [12, 33].

Moreover, the comparison of OpenSignal's Brazilian averages (§ 8.2) suggest that the difference between our results and speedtest averages is not a phenomenon exclusive to the MNO being studied but rather an observation that generalizes to the whole country, as the recorded averages are similar. We hypothesize this difference is partly explained by measurement bias in crowdsourced experiments. A subset of users, which are in many cases hard to reach and analyze (e.g. casual internet users, or with low familiarity with technology), can represent a large portion of a country's population. Additionally, there are contexts covered by our dataset on which users will likely not initiate a speedtest (e.g. while driving or under movement). We can try to remove the effect of those conditions: 18% of our measurements have handover; 30% have many devices at the tower; 41% have devices that are somewhat outdated (below 2Ghz clock speed). If we remove all these variables, the throughput median goes from 18 Mbps to 27 Mbps, a closer value to the speedtest report. The remaining difference suggests there are other environmental and situational factors that can impact throughput beyond the network's and client's inherent technological capacity, e.g., location and dispensible income.

**Takeaway:** *Although crowdsourced open-data sets provide valuable data, they have fundamental bias that cause them to overhead network performance.*

## 8.4 Mismatch Between Network Recommendations and Brazil

Given a lack of data, developers and designers often turn to standards and recommendations. Next, we identify mismatches between the recorded throughput and the reference values and guidelines currently available and the reality of today's infrastructure in Brazil. We also explain the bias which introduce these mismatches.

Firstly, there exists an assumption that developing regions have much worse connectivity, to the point that many common Internet use cases would not function properly. As we show in Section 7, post-pandemic infrastructure is fundamentally different from pre-pandemic with a dramatic decrease in 2G and 3G connections. As such, recommendations, e.g., Android developers guidelines [3]

[4]The speedtests report an average of 12.09 Mbps against 11.93 Mbps of our dataset.

from 2023, which claim that half of the world will use 2G connection are increasingly becoming antiquated. In fact, our dataset shows that only less than 0.1% of the connections use 2G.

The second bias is that the throughput of each connection technology is much lower than the theoretical reference values. For example, W3C's Network Information API [15] table of the maximum download speed for each connection technology shows 4G maximum download speed to be of 100 Mbps while we observe that only 8% of measurements of this study are above half this speed, at 50 Mbps. To illustrate the broader differences, Table 6 quantifies the gap between our measurements and the standards. In some cases, the technology (e.g., HSPA) has multiple releases and multiple vendor-specific enhancements, which increase the theoretical limit. Thus, the standards do not include the newer releases or vendor-specific enhances which creates another mismatch.

Finally, crowdsourced measurements, often used as a reference, are highly dependent on the specific methodology and context on which they were recorded. As we discussed in Section 8.3, the region on which the measurements were recorded, the biases in crowdsourced data and the measurement test methodology could all greatly influence the measured values. The complexity of large scale cellular infrastructures warrant further investigation on the factors that skew performance away from the reference values.

**Table 6: Theoretical limit (from [15]) vs. real-life averages of throughput, download (d) and upload (u) speeds in Mbps. Threshold is a strong signal value cuttoff.**

| Tech | Theoretical Limit | Strong Signal throughput | Poor Signal throughput | Threshold (dBm) |
|---|---|---|---|---|
| NR | 20000(d) | 158.3 (d) 45.1 (u) | 91.6 (d) 14.1 (u) | > -100 |
| LTE | 100 (d) | 26.89 (d) 18.82 (u) | 17.00 (d) 5.69 (u) | > -100 |
| HSPA+ | 42 (d) | 4.98 (d) 1.74 (u) | 3.34 (d) 0.99 (u) | > -85 |
| HSUPA | 14.4 (d) | 5.12 (d) 1.74 (u) | 3.76 (d) 1.10 (u) | > -85 |
| HSPA | 3.6 (d) | 5.15 (d) 1.99 (u) | 3.35 (d) 1.31 (u) | > -85 |
| UMTS | 2 (d) | 3.53 (d) 1.35 (u) | 2.98 (d) 1.01 (u) | > -85 |
| GPRS | 0.237 (d) | 1.78 (d) 1.34 (u) | 1.52 (d) 1.12 (u) | > -90 |

**Takeaway:** *There is a significant gap between recommendations from web organizations (Android, W3C) and the current infrastructure landscape of Brazil.*

## 9 Conclusion

In this paper, we analyze a dataset of measurements collected over 7.6 million client devices from a major MNO in Brazil. We take a deeper look at network and device variables and find that base station load, handovers and signal strength are among the top variables affecting the download and upload throughput alongside the connection technology being used. In addition, we compare network performance results with the numbers published by other network performance studies. We find that, while some numbers remain the same, the speedtest results could be overestimating the performance of the country's region and the same observation could apply to other regions. We also study the country's connectivity throughout 2023 and compare it with pre-pandemic levels, showing a significant reduction in the mobile digital divide.

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

## A Comparison with Previous Studies

Table 7 shows a summary of similar performance studies on cellular networks. We make the following observations: (a) the studies are either done on a selected subset of test devices with custom measurement software tested on different network conditions; or done using data from speedtest platforms or other services independent from an MNO; (b) the existing body of work lacks comprehensive studies done on Latin American networks and has insufficient studies on the global south; (c) Research tend to focus on a limited subset of network variables and conditions, which makes it difficult to generalize findings to the entire client base of a cellular network. Our work addresses these issues presenting an alternative measurement methodology and analysis.

## B Brazilian Network Characterization

Our dataset contains over 7 million real user devices distributed in 860 unique cell sectors. Next, we provide a general characterization to highlight diversity and heterogeneity, which allows us to draw general conclusions throughout the country.

**Client Distribution Mirrors Population:** In analyzing the distribution of mobile devices between towers, it is unsurprising that we observe that these devices are not uniformly distributed between stations. This naturally highlights differences in population densities. To validate this observation, we used the LAC information (i.e., Region) from each measurement session's base station to map the client to LAC: LAC is a two-digit number that identifies the 67 regions in Brazil. The LAC allows us to map devices to regions via the base station, which are assigned LAC codes. We observe that the relative number of measurements in each LAC matches its population density according to the latest population Census [19]. More information validating this observation is in Appendix C. Areas with high device density also have high population density. With a large sample size and similar geographical distribution, we are able to generalize insights on this dataset to the entire country.

**Cell Tower Characterization:** 4G is the most commonly used technology across the territory, having significant usage on 66% of the base stations. Although 3G is only used in around 40% of the cell towers, most 3G measurements are concentrated in a small number of base stations (15% of the total) with many devices recorded, which indicates that they happen on areas with high population density. This distribution implies that 3G is also heavily used in urban areas and not only on rural or remote areas. 5G measurements are present in very few base stations: 2.9% of the total. This is likely a result of the fact that the 5G infrastructure was not widely deployed at the time the data was collected because it requires different technology and has a different footprint than 3G/4G towers.

Next, we delve into connection technologies within each generation. Most (83%) connections use LTE and LTE with Carrier Aggregation (4G) technology. Next are HSPA+ (3G), and New Radio (5G) with 14% and 1%, respectively. The remaining are alternative 3G technologies (UMTS, HSDPA) and 2G measurements, which make up less than 0.02% of the total.

### B.1 5G Adoption

5G was utilized in just over 1% of total connections in 2023. 4G and 3G technologies remain mostly stable throughout 2023 as the most

**Table 7: Summary of related studies on cellular network performance.**

| Ref | Location | Dataset | Technology | Variables | Indicators |
|---|---|---|---|---|---|
| [32] | US, India, S. Korea, Europe | Volunteer speedtests on an internal Google Android application | 2G, 3G, 4G | technology, carrier, location,time of day, network path, signal strength | RTT, throughput, error rate and loss |
| [22] | US, Spain, France, Italy, Germany | Mobile measurements with selected devices and dedicated speedtest servers | 5G | MNO, channel bandwidth, MIMO configuration, MCS values, UE-tower distance | Physical-layer throughput and latency, bitrate, buffering, video quality |
| [31] | United States | Mobile measurements with selected devices | 4G, 5G | UE-Server distance, handover, signal strength, location, frequency band, ABR algorithms | Energy consumption, throughput, bitrate, RTT, page load time |
| [16] | United States | Mobile measurements under user movement with custom software and equipment | 4G, 5G | Frequency band, user mobility, handover, Connection technology | Latency, packet loss, bitrate, % dropped frames, Throughput |
| [53] | China | Set of measurements from a live streaming platform | 4G, 5G, | Connection technology, Handovers, carrier, access density, bitrate, network hops | Throughput, RTT, energy consumption |
| [49] | Finland | Mobile measurements from crowd-sourced speedtest platform | 2G, 3G, 4G | Technology, location, time of day, phone model, MNO, battery level, signal strength, handover | Throughput, link stability |
| Ours | Brazil | Real-user measurements from MNO Android app | 2G, 3G, 4G, 5G | Technology, time of day, phone model, battery level, signal strength, CPU, RAM, Number of attached devices, handover | Throughput |

common technologies. It is likely that both technologies will still be used by a significant share of the population in the following years. 5G is already more popular than 2G but is still rising timidly throughout the considered period. Compared to January of 2020, the country's network performance improved significantly. The average download and upload speed went from 11 Mbps and 5 Mbps, respectively, to 24 Mbps and 12 Mbps in 2023. The average download speed of 5G connections is 141 Mbps and upload speed is 36.2 Mbps, although some measurements record download speeds as high as 550 Mbps. The recorded averages are similar to 5G studies [28] recorded in other continents, and are enough to offer a good quality of service to network-intensive applications such as video streaming or downloading media.

## C   Client Distribution Mirrors Population

The Location Area Code is defined by the SIM card being used by the phone during the measurement session. We can infer in which LAC region each base station is located considering only the most common LAC from the measurement sessions at that base station. Therefore, we get (at a granularity of LAC region) the location of every device. In Figure 8 we plot devices and population in each LAC region. The Spearman correlation coefficient for these two variables give a value of 0.66; and fitting a regression line on those two variables gives an R-squared measure of 0.76, values that indicate a strong correlation between both variables.

## D   SHAP Values per Technology

In Figure 9 we see the feature importance ranking for New Radio. We note that signal strength and CPU clock speed are among the top contributors even when predicting a single connection technology. In other words, signal and clock speed are not entirely related to connection technology; they influence the output independently of it. The remaining variables retain their positions: signal strength, handover, CPU speed and number of devices are the top contributors for 5G. The same ordering happens when we analyze only LTE connections.

One of the possible explanations for the CPU clock speed effect on throughput is in Figure 10. We see the memory usage starts to

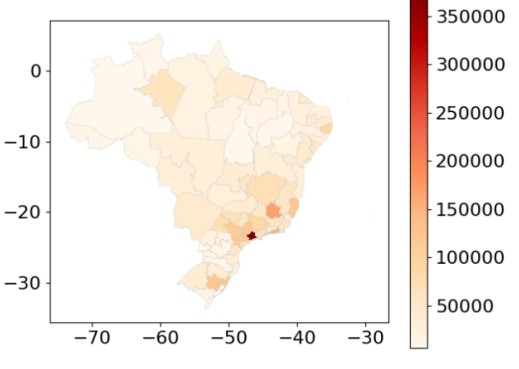

**(a) Device by LAC region**

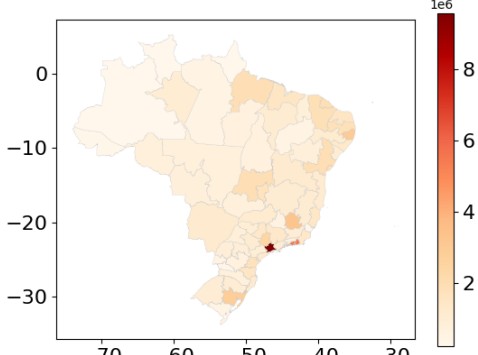

**(b) Population by LAC region**

**Figure 8: Device and population in each LAC region.**

negatively impact the prediction value at around 75%. This number is likely the threshold on which the cellphone cannot keep up network processing fast enough. Second, we see from the image that this scenario happens predominantly in CPU clock speeds

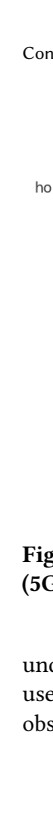
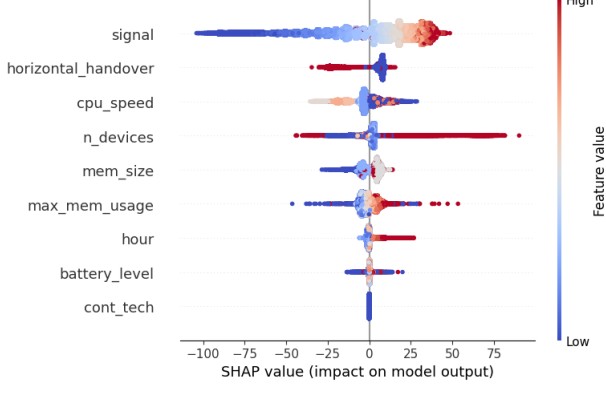

**Figure 9: Feature set ranked, considering only New Radio (5G) measurements.**

under 1,500 MHz, which may indicate that CPU clock speed is a useful proxy variable to indicate hardware (dis)advantage. This observation applies for both 4G and 5G connections.

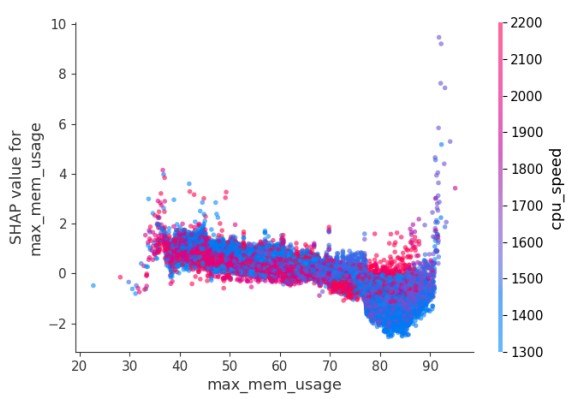

**Figure 10: Dependency plot of Max Mem Usage vs. CPU Speed.**

## E  Impact of Battery Level on Throughput

Battery level is a value between 1 and 100 that shows the device's battery level during the measurement session. The average recorded battery level is 57.97 and the standard deviation is 26.53. The measurements are fairly spread out across the range of possible values. There are slightly less measurements recorded on phones with very low battery, presumably because people leave their phones charged more often than not.

For unmatched data points, Spearman's Rank correlation shows $\rho = +0.008$ and $p = 10e - 21$. Although these values are statistically significant, the correlation coefficients are so low that it does not mean any effective correlation, in a practical sense. For matched pairs, Spearman shows $\rho = +0.034, p = 10e - 14$. Similarly, no significant correlation is found when neutralizing covariates. The

correlation coefficient for upload throughput is $\rho = +0.005$ and $p = 10e - 21$, almost identical values to download throughput.

Although there seems to be no correlations between battery level and throughput, many devices have a battery saving mode, usually activated when the device drops below 10% battery level. We can derive a binary metric from battery level that divides the data into measurements being made on low battery or otherwise. The average download throughput for low battery phones is 10.77 Mbps, and for high battery phones is 10.38 Mbps. Likewise, average upload throughput values measure 5.46 Mbps on low battery phones and 5.66 Mbps on high battery phones. Although values differ with a significant confidence interval on a t-test, this is not enough to conclude that battery level correlates to throughput.

***Takeaway:*** *Battery level did not exhibit a significant correlation with download or upload throughput. Likewise, Android's battery saving mode seems to have no correlation with measured throughput.*

