# OpenReview forum: "Hidden Impact of Hardware Technologies on Throughput: a Case Study on a Brazilian Mobile Web Network"
_ACM.org/TheWebConf/2025/Conference — WWW 2025 Poster_

### Official Review · Reviewer_YPNo · 2024-11-27

**Novelty:** 6
**Technical Quality:** 6

**Review:**

This paper presents a detailed analysis of mobile network performance in Brazil using a large-scale dataset from one of the country's major Mobile Network Operators (MNOs). The study examines how device characteristics and infrastructure factors impact throughput performance, with particular attention to differences between developed and developing regions. The paper is generally well-structured, with clear sections for methodology, analysis, and results. However, some technical terms (e.g., SHAP analysis) could be briefly explained for broader accessibility. The findings are highly relevant for network operators and application developers. By highlighting the discrepancies between global standards and local realities, the paper underscores the need for region-specific guidelines, especially in developing countries. This could drive more tailored mobile web optimizations and infrastructure investments.

## Pros

1. **Comprehensive Dataset**: The analysis uses an impressive dataset of 176 million measurement sessions from over 7 million devices, providing strong statistical significance.
2. **Novel Methodology**: The statistical ensemble combining traditional methods with machine learning approaches offers a robust analytical framework for understanding network performance factors.
3. **Temporal Analysis**: The study includes both pre- and post-pandemic data, enabling valuable insights into how mobile infrastructure evolved during this critical period.
4. **Practical Implications**: The findings have clear practical value for network operators, device manufacturers, and application developers working in developing regions.
5. **Bias Analysis**: The paper effectively demonstrates limitations in existing crowdsourced measurements and challenges assumptions about adoption patterns in developing regions.

## Cons

1. **Single Country Focus**: While Brazil is representative of developing regions, including data from other similar markets would strengthen generalizability.
2. **Limited iOS Data**: The study only includes Android devices, potentially missing insights from iOS users who may represent a different demographic.
3. **Network Provider Bias**: Data from a single MNO may not fully represent the entire Brazilian market.
4. **Technical Explanations**: Some methodologies, such as the matched-pairs approach, could be elaborated further for clarity.

## Overall Assessment
This paper makes a valuable contribution to understanding mobile network performance in developing regions, offering practical insights through a robust dataset and advanced analysis. While addressing biases and highlighting regional discrepancies, its focus on a single country, single MNO, and the exclusion of iOS data limit generalizability, leaving room for future improvement.

**Questions:**

How might your findings change if you included data from other MNOs in Brazil or neighboring countries and iOS devices were included in the analysis?

**Reviewer Confidence:**

1: The reviewer's evaluation is an educated guess

**Scope:**

4: The work is relevant to the Web and to the track, and is of broad interest to the community

---

### Official Review · Reviewer_XDKd · 2024-11-28

**Novelty:** 4
**Technical Quality:** 4

**Review:**

**Summary**

This study reveals the relationship between hardware technology and network performance by analyzing the implicit impact of hardware technology on throughput in Brazilian mobile networks. Using data provided by a Brazilian MNO, the study analyzes the performance differences and the reasons behind them from the point of view of signal strength, etc., and evaluates the changes in network performance before and after the epidemic.

**Pros**
+ Looks at an overview of the situation of mobile network systems in South America and the impact of factors such as infrastructure on the network.

+ The sample size involves the largest MNOs in South America, a large sample size with some breadth.

**Cons**

- Although changes before and after the epidemic are covered, the causes of this impact are not further analyzed in depth.

2. The data analysis only covers Brazil and does not indicate whether it is representative for South America.

3. For a single MNO, is the overall data affected by the characteristics of that MNO? The possibility of bias in the data set itself is not discussed in the text.

*Detailed comments for authors**
Thank you for submitting your research to WWW'25. This study focuses on the impact of hardware technology on Brazilian mobile web networks and examines changes before and after the epidemic. Although the study goes in an important and novel direction, there are still several areas that need to be improved, with the following recommendations.

***Presentation***
1, Explanation of abbreviations: For example, SHAP appears several times in the text, but initially there was no display of its full name and no explanation of its relevance.

2. The label in figure 5 (b) is too large and has obscured the presentation of the bar chart.

3. The meaning of the red dotted line in figure 4 (b) is unclear, and the respective meanings of 33% and 82% are not mentioned in context.

 ***Detailed comments***

1. On the dataset, is it possible to enumerate the distribution of phone brands so as to see if it is affected by a particular cell phone brand causing a bias.

2. Also, can the specific data window and specific operation of the dataset be reflected, or is this an ongoing testing project since 2020? Given that this is a dataset that has yielded 1.7 million pieces of data, is this a collaborative project with MNO? Could you describe in more detail the realization process?

3. Global vs. local variations: The Android Developer's Guide cited in Section 8.4 mentions that “half of the world's population will be using 2G connectivity”, which is based on global averages. However, Brazil, as a country in Latin America, is not necessarily comparable to the global average, and the guide does not mention any South American countries. This data discrepancy does not allow for the conclusion that the authors' perceptions are outdated.

4. Rigor of data citation: The statement “half of the world's users will use 2G connections” is taken from the contextualization of the developer guide and is not based on rigorous data or research findings. This guidance is written as a guide for developers to ensure application compatibility and does not necessarily reflect actual network usage distribution and perceptions.

5. whether the discussion of the results could be more in-depth, for example:
Regarding the relationship between signal strength and throughput, the practical application implications of signal optimization are not discussed despite the use of matched pair analysis.
The similarity between 4G and 3G throughput was mentioned but the reasons for this were not discussed in depth.
In the case of throughput and technology improvement before and after the epidemic in question, there is no discussion of its specific causes based on the epidemic; the factors visible are 5G development and LTE adoption, and it is suggested that epidemic-based factors be added to the discussion.

6. When discussing the relationship between throughput and signal strength, the authors used figure 4(c) and table 4, to analyze that throughput is correlated with both signal strength and signal generation. However, as seen in the figure, there is a fairly significant crossover between 3G and 2G at -80dBm, and 2G clearly shows a steeper trend compared to other technology generations, can the authors explain this further?

**Questions:**

1. In response to the questions about the data collection methodology and time window, could you add more details about the sampling strategy and how it works exactly?

2. If specificity is based on regional differences, can it be argued that the Brazilian situation is representative for South America or even the South, given that Brazilian throughput is approaching that of Southern European countries?

**Reviewer Confidence:**

2: The reviewer is willing to defend the evaluation, but it is likely that the reviewer did not understand parts of the paper

**Scope:**

3: The work is somewhat relevant to the Web and to the track, and is of narrow interest to a sub-community

---

### Official Review · Reviewer_9BEu · 2024-12-01

**Novelty:** 1
**Technical Quality:** 1

**Review:**

This work is not in my area at all.

**Questions:**

-

**Reviewer Confidence:**

1: The reviewer's evaluation is an educated guess

**Scope:**

1: The work is irrelevant to the Web

---

### Official Review · Reviewer_cefH · 2024-12-02

**Novelty:** 5
**Technical Quality:** 5

**Review:**

This paper presents a study that investigates the impact that the characteristics of hardware have on the network throughput (and other application-level measures of performance) of mobile devices. Importantly, the study is performed using data collected from one of the largest Brazilian mobile network operators. The paper highlights differences between this dataset and the assumptions that are held on the basis of data gathered from North America and Europe, indicating the limitations of approaches that do not consider, or underrepresent, the global south.

# Strengths

- The data and analysis are useful to the community, especially considering that such datasets have typically been provided for North America and Europe, with less representation from other regions around the world. A complete view of deployments throughout the world is necessary to ensure that standards and implementation work matches real-world experience.
- The above is demonstrated by the clear and concise findings that are presented; there are implications for standards bodies (e.g., the W3C) whose recommendations are based upon network conditions that might not exist for large numbers of users.
- The paper is well written.

# Limitations

- The comparison between the results that are described here and those obtained from the Ookla Speed Test platform are fairly significant, but the analysis is fairly narrow.

**Questions:**

- The paper notes that the measurement tool is included inside the network operators mobile app -- what triggers a measurement?
- Section 8.4 gives a couple of examples where standards (from the W3C) and developer documentation (from Android) are at odds with the results that are described -- is there any way to be more systematic about identifying such instances?
- What is the reason for splitting the dataset into pre- and post-pandemic, other than the existence of the pandemic? Did the pandemic drive investment in the network in some way? Did it change how people use the network?

**Reviewer Confidence:**

2: The reviewer is willing to defend the evaluation, but it is likely that the reviewer did not understand parts of the paper

**Scope:**

3: The work is somewhat relevant to the Web and to the track, and is of narrow interest to a sub-community